# FastFlow: Accelerating The Generative Flow Matching Models with Bandit Inference

**Divya Jyoti Bajpai[1], Dhruv Bhardwaj[2], Soumya Roy[2], Tejas Duseja[2],**
**Harsh Agarwal[2], Aashay Sandansing[1], Manjesh K. Hanawal[1]**

[1]Indian Institute of Technology Bombay
[2]Amazon

```
{divyajyoti.bajpai, 23d1594, mhanawal}@iitb.ac.in
{dhruvbrd, dusejat, hragarwl}@amazon.com
meetsoumyaroy@gmail.com
```

## Abstract

Flow-matching models deliver state-of-the-art fidelity in image and video generation, but the inherent sequential denoising process renders them slower. Existing acceleration methods like distillation, trajectory truncation, and consistency approaches are static, require retraining, and often fail to generalize across tasks. We propose FastFlow, a plug-and-play adaptive inference framework that accelerates generation in flow matching models. FastFlow identifies denoising steps that produce only minor adjustments to the denoising path and approximates them without using the full neural network models used for velocity predictions. The approximation utilizes finite-difference velocity estimates from prior predictions to efficiently extrapolate future states, enabling faster advancements along the denoising path at zero compute cost. This enables skipping computation at intermediary steps. We model the decision of how many steps to safely skip before requiring a full model computation as a multi-armed bandit problem. The bandit learns the optimal skips to balance speed with performance. FastFlow integrates seamlessly with existing pipelines and generalizes across image generation, video generation, and editing tasks. Experiments demonstrate a speedup of over $2.6\times$ while maintaining high-quality outputs. The source code for this work can be found at `https://github.com/Div290/FastFlow`.

## 1 Introduction

Recently, **flow-matching (FM) models** Lipman et al. (2022) have emerged as an effective approach for visual generation, offering both high fidelity and computational efficiency. By learning continuous vector fields that transport simple distributions to complex data distributions, they generate samples along smooth, iterative trajectories. Unlike diffusion models Croitoru et al. (2023), FM achieves faster convergence and fewer sampling steps while maintaining comparable or better perceptual quality. This framework allows precise control over the generative process, where increasing the number of flow integration steps typically improves perceptual quality in both images and videos. Despite these advances, **inference speed remains a major bottleneck** Yan et al. (2024); Davtyan et al. (2025) due to the several reverse denoising steps that are performed sequentially. As model sizes grow and generation tasks demand higher resolutions or longer video durations, the computational cost becomes prohibitive, resulting in substantial latency during inference.

Several acceleration strategies—such as *distillation* Luhman & Luhman (2021); Yan et al. (2024); Kornilov et al. (2024), *trajectory truncation* Dhariwal & Nichol (2021); Lu et al. (2022); Liu et al. (2025a), and *consistency training* Yang et al. (2024); Zhang & Zhou (2025); Dao et al. (2025)—have been proposed. While effective, these approaches have limitations: they require additional training phases, rely on large-scale data, and incur non-trivial computational overhead. Moreover, they apply a uniform inference schedule across all inputs, overlooking the fact that some samples may converge

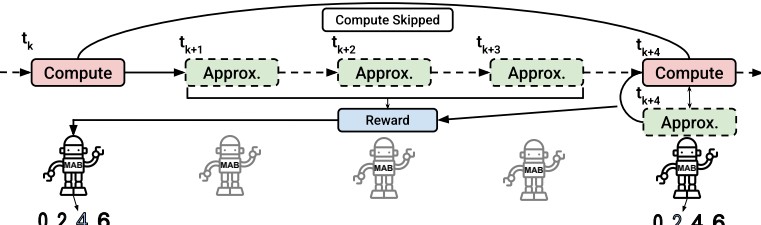

Figure 1: Overview of our method. At each step, the multi-armed bandit (MAB) selects the number of steps to approximate the trajectory. The bandit receives a reward proportional to the number of steps successfully approximated, while deviations from the computed velocity incur a penalty. This adaptive strategy allows the model to balance efficiency and accuracy across the trajectory.

with fewer steps, while others require longer trajectories to maintain fidelity. This one-size-fits-all design leads to inefficiencies, as many intermediate steps contribute little to the final quality.

In this work, we propose a **novel adaptive inference framework** that reduces cost by *approximating redundant intermediate denoising steps* instead of fully computing them. Our approach builds on the observation that flow-matching models often follow approximately linear denoising trajectories Lipman et al. (2022) as they are trained to follow linear paths. Leveraging this property, we approximate future states using **Taylor series expansions** and *local velocity estimates* derived from the model dynamics, thereby reducing the number of expensive forward passes. We show that our approximation is sound by establishing a theoretical bound on the deviation of the final state of the approximate trajectory from that of the full model. When approximation fails to maintain fidelity, the system reverts to full model predictions. The central challenge, therefore, is to determine when approximation suffices and when precise computation is necessary.

We address this challenge by formulating the decision process as a **multi-armed bandit (MAB) problem**. At each step, a bandit adaptively selects how many future steps can be approximated before requiring the next full model evaluation. Each arm corresponds to the number of steps to approximate, and a subsequent full evaluation provides feedback to assess approximation accuracy. The reward balances two competing objectives: (i) reducing computational cost by skipping evaluations, and (ii) limiting deviation from the true model trajectory. This adaptation allows the system to adjust inference complexity on a per-sample basis, **learning over time** when approximation is reliable and when exact prediction is necessary. Fig. 1 illustrates our method (details in Section 3.2).

In existing caching-based acceleration approaches, **TeaCache** Liu et al. (2025a), caches residuals and re-uses them at later inference steps. However, they use a hand-crafted relative-$L1$ distance-based criteria to decide if a cache can be reused. In our experiments, we also observed that when a specific speedup (e.g., $2\times$) is required, TeaCache unintentionally ends up with a fixed caching schedule across generations, consistently skipping the same subset of timesteps regardless of input complexity. Finally, TeaCache relies on handcrafted polynomial fitting of noisy inputs, which typically requires prior model- or task-specific knowledge to perform optimally (see Appendix A.1).

Our approach is both **efficient** and **adaptive**: at every timestep, the bandit dynamically learns to minimize redundancy by adapting to the complexity of the data distribution. Unlike prior acceleration strategies, our method introduces *zero retraining overhead*, requires no auxiliary networks, and integrates seamlessly as a true **plug-and-play** solution. In summary, our key contributions are:

- We propose FastFlow, a method to accelerate visual generation skipping redundant denoising steps, while using a simple Euler-solver for flow, and a first-order Taylor series expansion for velocities. The trade-off of speed v/s error is set up as a Multi-Armed Bandit (MAB) objective, enabling the model to dynamically learn when full computation is necessary.

- We establish a theoretical bound on the deviation of the final state from the approximated trajectories with that obtained by the full model (see Thm. 3.1).

- Our framework is model-agnostic, requires no retraining or auxiliary networks, and can be seamlessly integrated into existing flow-matching pipelines, making it a practical and general solution for faster visual generation.

- Extensive experiments across image generation, video generation, and image editing demonstrate more than $2.6\times$ speedup while maintaining generation quality, showing that our method achieves acceleration without sacrificing fidelity.

## 2 RELATED WORKS

Recently, Flow Matching Lipman et al. (2022); Dao et al. (2023); Labs et al. (2025); Deng et al. (2025) has gained prominence as a strong counterpart to diffusion models Croitoru et al. (2023); Xing et al. (2024); Yang et al. (2023); Zhu et al. (2023), since it establishes a deterministic correspondence between random noise and data. This benefits applications such as image inversion Deng et al. (2024), editing Wang et al. (2024), and video synthesis Kong et al. (2024); Wan et al. (2025), where reduced randomness leads to faster sampling with fewer neural evaluations. Multimodal variants, like FlowTok He et al. (2025), compress text and images into a joint token space to improve inference speed, and large-scale systems like FLUX.1 Labs et al. (2025) demonstrate that flows can approach the performance of diffusion models at low compute cost. In video domains, Pyramidal Flow Matching Jin et al. (2024) cuts down complexity using hierarchical generation.

Nonetheless, the reliance on iterative sampling continues to hinder real-time deployment. To mitigate this, most acceleration work has involved retraining-based schemes Lee et al. (2023); Bartosh et al. (2024). Knowledge distillation methods Luhman & Luhman (2021); Song et al. (2023); Liu et al. (2022b); Kornilov et al. (2024); Salimans & Ho (2022), including InstaFlow Liu et al. (2023), LeDiFlow Zwick et al. (2025), and Diff2Flow Schusterbauer et al. (2025), leverage diffusion priors for one- or few-step generation, while PeRFlow Yan et al. (2024) simplifies trajectories via piecewise rectified flows. Sampling-acceleration approaches Dhariwal & Nichol (2021); Lu et al. (2022); Shaul et al. (2023), such as TeaCache Liu et al. (2025a), skip redundant steps using timestep embeddings, whereas consistency-based methods Yang et al. (2024); Zhang & Zhou (2025); Haber et al. (2025); Dao et al. (2025) combine adversarial and consistency objectives for high-fidelity few-step synthesis. Additionally, higher-order pseudo-numerical solvers, as in PNDM Liu et al. (2022a), improve the speed–quality tradeoff in diffusion inference.

While existing methods remain static and often struggle to generalize across different models, tasks, and datasets, FastFlow offers a universally compatible solution for any FM-based model. It dynamically adapts, identifying and skipping redundant steps based on the incoming data distribution, while the finite-difference approximation boosts efficiency to achieve significant speedups. Remarkably, FastFlow is training-free and incurs negligible computational overhead, making it both practical and highly effective in real-world scenarios.

## 3 METHODOLOGY

In this section, we provide details of our method, starting with a detailed description of the flow matching models and then detailing our application to the flow matching models.

### 3.1 FLOW MATCHING OVERVIEW

Consider two probability densities $\pi_0$ and $\pi_1$ on $\mathbb{R}^d$, representing the source and target distributions. **Flow Matching (FM)** seeks to learn a deterministic, time-continuous flow that transports samples from $\pi_0$ to $\pi_1$, governed by an ordinary differential equation (ODE). Formally, FM introduces a time-dependent velocity field $v : \mathbb{R}^d \times [0, 1] \rightarrow \mathbb{R}^d$, giving rise to the initial value problem

$$\frac{dx_t}{dt} = v(x_t, t), \quad x_0 \sim \pi_0, \quad t \in [0, 1]. \tag{1}$$

Here, $x_t \in \mathbb{R}^d$ denotes the sample state at time $t$, while the velocity field $v(x_t, t)$, typically parameterized by a neural network, is optimized such that the terminal distribution at $t = 1$ matches $\pi_1$. This ODE defines a flow map $\Phi_t(x_0)$ that evolves samples along continuous trajectories, with $\Phi_1(x_0) \sim \pi_1$. The central task in FM is thus to learn a velocity field $v(x_t, t)$ that realizes this transport.

**Inference:** As closed-form solutions for $x_t$ are generally unavailable for learned velocity fields, numerical solvers are required. FM most often employs the **forward Euler method** for its simplicity

and efficiency. The interval $[0, 1]$ is discretized into steps $\{t_0 = 0, t_1, \ldots, t_K = 1\}$, possibly with non-uniform intervals. Starting from $x_0 \sim \pi_0$, the trajectory is advanced as

$$x_{t_{k+1}} = x_{t_k} + \Delta t_k \cdot v(x_{t_k}, t_k), \quad \Delta t_k = t_{k+1} - t_k. \tag{2}$$

This discretization approximates the continuous flow with a finite sequence of updates, where each step moves the sample in the direction given by the velocity field. Owing to its low computational cost and suitability for parallel hardware, Euler's method remains the default choice in most FM implementations.

For our method, we need to find all the redundant denoising steps, as there will be many due to straight line trajectories that are learned during training and then replace them using some good approximation of the true model predictions.

## 3.2 OUR METHOD

**Approximating velocity.** To accelerate sampling, we look at a simple mechanism to approximate velocities at different timesteps, instead of re-computing from the model. Using the first-order Taylor series expansion for $x_{t+\Delta t}$ and taking a time derivative results in:

$$v(x_{t+\Delta t}, t + \Delta t) := \frac{dx_{t+\Delta t}}{dt} = v(x_t, t) + \Delta t \cdot \frac{dv(x_t, t)}{dt} \tag{3}$$

A natural direction for approximation is to use the most recent velocity estimate computed from the model assuming $\frac{dv(x_t, t)}{dt} \to 0$. In previous works, this is accompanied by a static criteria to decide whether re-computation from the model is necessary.

However, we find that even in the regions where velocity seems to be smooth and linear, it makes minor adjustments, ignoring those leads to accumulated errors during generation (see Figure 4), making above strategy overly simplistic for aggressive skipping.

Instead of re-using the same velocity estimate, we update it using **finite-difference approximation** of Eq. 3, utilizing the past velocity estimates. We write this approximation at discrete time steps as follows (where $p < k$):

$$v(x_{k+1}, t_{k+1}) \approx v(x_k, t_k) + \Delta t_k \cdot \frac{v(x_{t_k}, t_k) - v(x_{t_p}, t_p)}{t_k - t_p}. \tag{4}$$

Below we establish a bound on the deviations in the flow value of the approximated velocity estimates while skipping a set of time-steps during inference with that obtained by the full model.

**Theorem 3.1.** *Let $\{x_{t_k}^{\text{true}}\}$ denote the trajectory obtained using the exact velocity field with the forward Euler method, and let $\{x_{t_k}^{\text{approx}}\}$ be the trajectory where velocity evaluations are skipped at a subset of steps $\mathcal{S} \subseteq \{0, \ldots, T-1\}$ and are instead approximated, for simplicity, via a first-order Taylor expansion in time. Under assumptions of smoothness of velocity field, the cumulative error in the final state after $T$ steps with a uniform step size $\Delta t = 1/T$ is bounded by:*

$$e_T := \|x_{t_T}^{\text{approx}} - x_{t_T}^{\text{true}}\| = \mathcal{O}\left(\frac{|\mathcal{S}|}{T^3}\right).$$

The proof of this theorem can be found in Appendix B. This result shows that the final error grows linearly with the number of steps skipped and thus provides a formal guarantee on the stability of our approximation scheme.

**Deciding Redundant Steps.** A central challenge in our framework lies in determining *when* to perform a model evaluation versus *when* to rely on an approximation. Since each approximation inevitably introduces error, uncontrolled propagation may cause the trajectory to deviate significantly from the true dynamics. Moreover, the tolerance to approximation errors can vary across samples of different complexity, implying that the decision criterion must adapt dynamically to the evolving data distribution. Thus, we cast the problem of detecting redundant steps as an *online sequential decision-making problem*, formalized via the Multi-Armed Bandit (MAB) framework.

In an MAB setup, an agent iteratively selects actions from a finite set, aiming to maximize cumulative reward while balancing *exploration* of uncertain actions and *exploitation* of actions known to

yield high rewards. At timestep $t_k$, let $\mathcal{A}_{t_k}$ denote the action set, where each action $\alpha_{t_k} \in \mathcal{A}_{t_k}$ corresponds to skipping $\alpha_t$ steps before the next model evaluation. A separate bandit is instantiated at each timestep, learning an adaptive policy for choosing $\alpha_t$ based on approximation performance.

Let $v(x_{t_k}, t_k)$ denote the true model velocity, $\hat{v}(x_{t_k}, t_k)$ its approximation under the chosen skip strategy, and $\ell(\cdot, \cdot)$ is a discrepancy measure (e.g., mean-squared error). We define the reward associated with action $\alpha_{t_k}$ as

$$r(\alpha_{t_k}) = \mu \cdot \alpha_{t_k} - \ell\big(\hat{v}(x_{t_k}, t_k), v(x_{t_k}, t_k)\big), \tag{5}$$

The scalar $\mu > 0$ balances the trade-off between efficiency (favoring larger $\alpha_t$) and accuracy (penalizing deviation from the true velocity).

This reward structure formalizes the intuition that skipping more steps accelerates inference, but incurs a penalty proportional to the local error. The MAB objective then becomes $\max_\pi \mathbb{E}\left[\sum_{t=1}^{T} r(\alpha_{t_k})\right]$, where $\pi$ denotes the adaptive policy that maps history of past rewards and actions to the choice of $\alpha_t$. By construction, the optimal policy $\pi^\star$ learns to exploit redundancies in locally smooth regions of the trajectory while reverting to exact model evaluations in regions of high curvature or instability.

**Algorithm.** Algorithm 1 presents the pseudo-code of FastFlow. The procedure begins with initialization: we specify the timestep grid, the velocity prediction model $\mathcal{M}$, the action sets $\mathcal{A}_{t_k}$ available to each bandit $\mathcal{B}_{t_k}$, and the trade-off parameter $\mu$. Each bandit is then initialized from a full generation using the first prompt, ensuring that each action is at least played once.

At inference time, when the trajectory reaches a state $x_{t_k}$, the corresponding bandit $\mathcal{B}_{t_k}$ selects a skip length $m := \alpha_{t_k}$ via an upper-confidence bound strategy (line 5). This choice reflects a balance between exploration of new skip patterns and exploitation of those that have yielded high reward. The trajectory then advances $m$ steps using finite-difference extrapolation, followed by an exact evaluation of $\mathcal{M}$ at the terminal point.

The reward couples efficiency with reliability: longer skips are encouraged by the term $\mu \cdot \alpha_{t_k}$, but this gain is counterbalanced by a velocity mismatch loss that anchors accuracy. Concretely, if $\alpha_{t_k} = m$, the extrapolated velocity $\hat{v}(x_{t_{k+m+1}}, t_{k+m+1})$ is contrasted with the true velocity $v(x_{t_{k+m+1}}, t_{k+m+1})$. This loss is crucial, as it directly measures the drift introduced by approximation: even small velocity errors accumulate along the trajectory, so penalizing the mismatch ensures stability. By continually updating bandit statistics under this trade-off, FastFlow adapts its policy across timesteps, recomputing when approximation would deviate significantly, while exploiting skips where the loss remains small.

**Computational Complexity of Bandits:** FastFlow employs multi-armed bandits (MABs) to determine the number of steps to skip and approximate. MABs are computationally lightweight, adding negligible overhead, as they only maintain a list of rewards computed as in line 5 of Algorithm 1. This efficiency is further confirmed empirically in our experiments.

**FastFlow and Linear multi-step solvers:** We reformulate an $m$ skip due to our method, on top of a Euler solver as a multi-step update in Appendix C.

## 4 EXPERIMENTS

We evaluate our approach across text-to-image generation, image editing, and text-to-video generation. Below we describe the datasets used in each setting.

**Datasets:** We use the **GenEval** benchmark Ghosh et al. (2023), a collection of 553 prompts designed to evaluate compositional reasoning in text-to-image generation. The prompts are organized to probe key abilities such as *object occurrence*, *spatial relations*, *color binding*, and *numerical consistency*, making GenEval a widely adopted standard for testing fine-grained semantic alignment.

For image editing, we adopt the **GEdit** benchmark Liu et al. (2025b), which comprises 606 real-world editing instructions in English. The instructions span a broad spectrum of operations—including *object manipulation*, *color changes*, *layout adjustments*, and *stylization*—allowing systematic evaluation of both localized edits and global scene transformations.

---

**Algorithm 1** FastFlow: Bandit-driven approach for accelerated Flow Matching inference

---

**Require:** $x_{t_0}$: Initial flow state
    $\{t_0, t_1, \ldots, t_T\}$: Timesteps
    $\mathcal{M}$: Velocity model
    $\{\mathcal{B}_{t_k}\}_{k=0}^{T-1}$: Bandit agents
    $\{\mathcal{A}_{t_k}\}$: Action sets for bandit agents
    $\mu, \gamma = 2.0$: Trade-off parameter, Exploration constant.
    $k$: Time index (loop variable)
    $p$: Time index of most recent (actual) velocity evaluation
    $m$: Skip length (in number of time indices)

1: Initialize the mean arm rewards $Q$ and arm counts $N$ for bandit agents $\{\mathcal{B}_{t_k}\}_{k=0}^{T-1}$ using the first prompt.
2: Compute initial velocities $v(x_{t_0}, t_0), v(x_{t_1}, t_1) \leftarrow \mathcal{M}(x_{t_0}, t_0), \mathcal{M}(x_{t_1}, t_1)$ and set $p \leftarrow 0$.
3: $x_{t_2} \leftarrow x_{t_1} + v(x_{t_1}, t_1) \cdot (t_1 - t_0)$.
4: $k \leftarrow 2$
5: **while** $k \leq T - 1$ **do**
6:     $n \leftarrow$ number of time $\mathcal{B}_{t_k}$ is invoked.
7:     Bandit $\mathcal{B}_{t_k}$ selects skip length $m := \alpha_{t_k} \leftarrow \arg\max_{\alpha \in \mathcal{A}_{t_k}} \left[ Q(\alpha) + \gamma\sqrt{\frac{\ln n}{N(\alpha)}} \right]$.
8:     Define $\hat{v}(t_k, t_p, \Delta t) := v(x_{t_k}, t_k) + \Delta t \cdot \frac{v(x_{t_k}, t_k) - v(x_{t_p}, t_p)}{t_k - t_p}; (p < k)$.
9:     **if** $m > 0$ **then**
10:       $\Delta t_{k+m,k} \leftarrow t_{k+m} - t_k$.
11:       $x_{t_{k+m}} \leftarrow x_{t_k} + \hat{v}(t_k, t_p, \Delta t_{k+m,k}) \cdot \Delta t_{k+m,k}$.
12:     **end if**
13:     $\Delta t_{k+m+1,k+m} \leftarrow (t_{k+m+1} - t_{k+m})$.
14:     $x_{t_{k+m+1}} \leftarrow x_{t_{k+m}} + \hat{v}(t_k, t_p, \Delta t_{k+m+1,k+m}) \cdot \Delta t_{k+m+1,k+m}$.
15:     $v(x_{t_{k+m+1}}, t_{k+m+1}) = \mathcal{M}(x_{t_{k+m+1}}, t_{k+m+1})$.
16:     Compute reward: $r(\alpha_{t_k}) = \mu \cdot \alpha_{t_k} - \ell\big(\hat{v}(t_k, t_p, \Delta t_{k+m+1,k+m}), v(x_{t_{k+m+1}}, t_{k+m+1})\big)$.
17:     Update bandit $\mathcal{B}_{t_k}$ statistics: $N(\alpha_{t_k}) \leftarrow N(\alpha_{t_k}) + 1, \quad Q(\alpha_{t_k}) \leftarrow \frac{\sum_{j=1}^{k} r(\alpha_j) \mathbb{1}_{\{\alpha_j = \alpha_{t_k}\}}}{N(\alpha_{t_k})}$.
18:     $p \leftarrow k; \quad k \leftarrow k + m$
19: **end while**
**Ensure:** Final trajectory $\{x_{t_k}\}_{k=0}^{T}$.

---

To measure temporal and multimodal consistency in video generation, we use a subset of **VBench** dataset . We construct a representative evaluation set by sampling 80 prompts, uniformly selecting 5 from each of the 16 dimensions defined by the benchmark. This ensures balanced coverage across diverse factors such as *motion dynamics*, *object persistence*, *camera control*, and *scene composition*, yielding a challenging yet comprehensive testbed for generative video models.

**Baselines:** We compare our method against the following baselines:

**Full Generation:** The standard sampling procedure, where the model executes the complete denoising trajectory without acceleration. This serves as the fidelity upper bound and the reference point for all accelerated methods.
**TeaCache:** TeaCache accelerates generation by caching intermediate representations and reusing them across timesteps. This eliminates redundant computation and reduces inference time, though fidelity can degrade due to approximations introduced in cached states.
**InstaFlow:** A flow-matching–based sampler trained for ultra-fast generation (down to a single step) on the Stable-Diffusion-v1.5 model. While highly efficient, it sacrifices fidelity compared to full sampling. We evaluate InstaFlow using the released Stable-Diffusion-v1.5 weights.
**PeRFlow (Piecewise Rectified Flow):** PeRFlow Yan et al. (2024) straightens the diffusion trajectory via piecewise-linear rectification over segmented timesteps, enabling few-step generation with favorable quality–efficiency tradeoffs. We report using the official Stable-Diffusion-XL checkpoints.
**Ours:** Our approach accelerates inference by selectively approximating redundant steps. A Multi-Armed Bandit dynamically decides where to apply approximation, balancing efficiency with fidelity.

| Method | SO | TO | CT | CL | ATTR | PO | Overall ↑ | CLIPIQA ↑ | Spd. ↑ | Lat. ↓ |
|--------|----|----|----|----|------|----|-----------|-----------|--------|--------|
| **Full Model** | | | | | | | | | | |
| Full 50 | 0.99 | 0.90 | 0.81 | 0.85 | 0.59 | 0.54 | **0.78** | **0.85** | 1.00× | 36.2 |
| Full 25 | 0.99 | 0.91 | 0.78 | 0.84 | 0.62 | 0.51 | 0.77 | 0.82 | 2.00× | 19.5 |
| Full 10 | 0.99 | 0.88 | 0.68 | 0.84 | 0.56 | 0.48 | 0.74 | 0.75 | 5.00× | 07.3 |
| **Static Speedup Methods** | | | | | | | | | | |
| InstaFlow | 0.86 | 0.20 | 0.21 | 0.66 | 0.04 | 0.02 | 0.33 | 0.74 | **50.0×** | **01.5** |
| PerFlow | 0.99 | 0.79 | 0.44 | 0.85 | 0.25 | 0.15 | 0.58 | 0.80 | 5.00× | 08.2 |
| Teacache | 0.99 | 0.89 | 0.78 | 0.83 | 0.58 | 0.52 | 0.76 | 0.80 | 1.85× | 20.6 |
| **Ours (FastFlow)** | | | | | | | | | | |
| FastFlow-50 | 0.99 | 0.91 | 0.80 | 0.86 | 0.63 | 0.51 | **0.78** | 0.83 | 2.65× | 13.7 |
| FastFlow-25 | 0.99 | 0.91 | 0.76 | 0.84 | 0.59 | 0.50 | 0.77 | 0.80 | 4.54× | 08.6 |
| FastFlow-10 | 0.98 | 0.84 | 0.65 | 0.84 | 0.54 | 0.47 | 0.72 | 0.73 | 7.14× | 05.5 |

Table 1: Comparison of flow-matching acceleration methods. SO: Single Object, TO: Two Object, CT: Counting, CL: Color, ATTR: Color Attribute, PO: Position, Overall: Overall score, CLIPIQA: Perceptual Image Quality. Speedup is relative to full-50 step generation. Latency is average inference time per image (s).

| Method | SO | TO | CO | CL | ATTR | PO | Overall ↑ | CLIPIQA ↑ | Spd. ↑ | Lat. ↓ |
|--------|----|----|----|----|------|----|-----------|-----------|--------|--------|
| **Full Model** | | | | | | | | | | |
| Full 50 | 0.98 | 0.79 | 0.73 | 0.78 | 0.44 | 0.21 | **0.65** | **0.84** | 1.00× | 33.8 |
| Full 25 | 0.98 | 0.78 | 0.71 | 0.76 | 0.43 | 0.18 | 0.64 | 0.80 | 2.00× | 17.5 |
| Full 10 | 0.97 | 0.66 | 0.59 | 0.67 | 0.41 | 0.15 | 0.57 | 0.60 | 5.00× | 07.1 |
| **TeaCache** | | | | | | | | | | |
| TeaCache-50 | 0.98 | 0.79 | 0.71 | 0.76 | 0.43 | **0.21** | 0.64 | 0.80 | 1.91× | 18.3 |
| TeaCache-25 | 0.97 | 0.76 | 0.70 | 0.74 | 0.43 | 0.17 | 0.62 | 0.78 | 3.45× | 10.3 |
| **FlowFast (Ours)** | | | | | | | | | | |
| FlowFast 50 | 0.97 | 0.78 | 0.72 | 0.77 | 0.44 | 0.20 | 0.64 | 0.82 | 2.57× | 13.9 |
| FlowFast 25 | 0.97 | 0.78 | 0.71 | 0.75 | 0.42 | 0.18 | 0.63 | 0.79 | 4.21× | 08.5 |
| FlowFast 10 | 0.95 | 0.64 | 0.54 | 0.66 | 0.40 | 0.14 | 0.55 | 0.57 | **7.59×** | **05.2** |

Table 2: **Comparison of Full model, TeaCache, and FlowFast (ours).** Best values in each column are **bolded**. FlowFast achieves significantly better performance-efficiency trade-offs while maintaining competitive accuracy and perceptual quality.

For all baselines, we adopt the official hyperparameters provided in their codebases. In Table 1, Tea-Cache is applied is as released, and in Figure 2, we further evaluate it across timesteps to emphasise its plug-and-play flexibility. An ablation over $\mu$ is given in Figure 7.

**Models.** To demonstrate the versatility of our approach, we evaluate across multiple state-of-the-art models: for image generation, BAGEL, Flux-Kontext, and PeRFlow; for image editing, BAGEL, Flux-Kontext, and Step-1X-Edit; and for video generation, HunyuanVideo.

**Hyperparameters.** We consider two key hyperparameters. (i) *Arm set:* Each arm represents the number of steps to skip. Since the feasible skip length naturally decreases as the remaining steps shrink, we design the arm set adaptively with respect to the current generation step. For fairness, the arm set is kept fixed across models and tasks, and updated only when the generation horizon changes. (ii) *Error scaling factor $\mu$:* To normalize rewards, we define $\mu = \frac{\max_t \text{MSE}(\hat{v}_t, v_t)}{\text{total steps}}$, where the maximum MSE is estimated from the first full generation pass. This choice rescales error values to the same order as step counts, ensuring stable bandit updates while explicitly encoding the trade-off between efficiency (fewer steps) and fidelity (lower error). We run the experiments on a single NVIDIA A100 GPU.

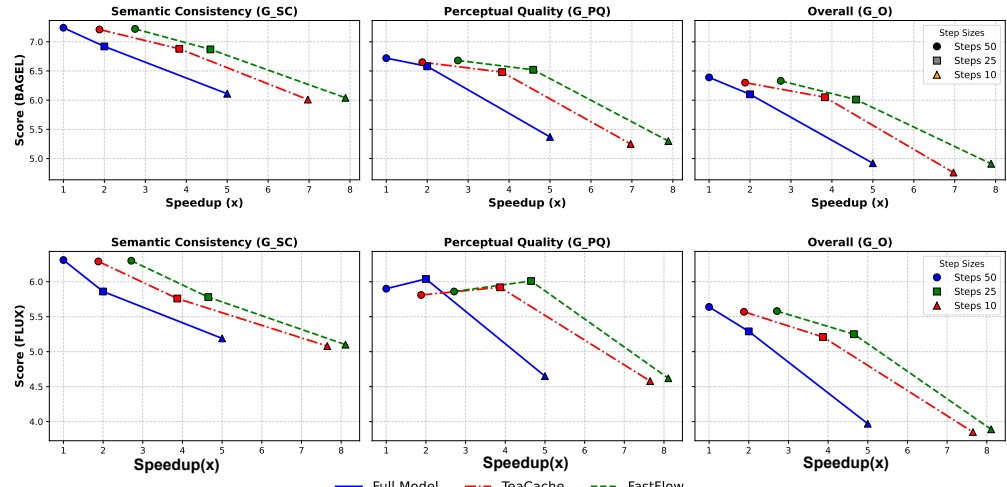

Figure 2: Comparison of edit quality across two models: BAGEL and FLUX. Each subfigure reports semantic consistency (G_SC), perceptual quality (G_PQ), and overall score (G_O) versus speedup.

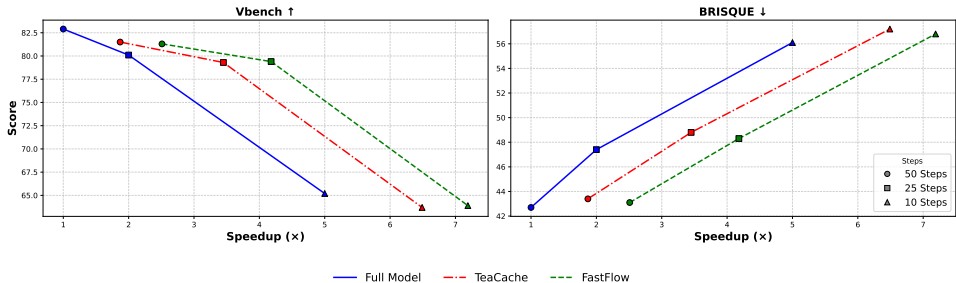

Figure 3: Comparison of the Video generation for the HunYuanVideo model. We report the VBench score an the BRISQUE metric for the quality of frames generated.

## 4.1 RESULTS

**Image Generation:** Tables 1 and 2 present a comprehensive evaluation of FastFlow against existing baselines across multiple dimensions, including single-object (SO) and two-object (TO) generation, compositional accuracy, and object positioning. While semantic correctness is important, perceptual quality is equally critical; to capture this, we report CLIPIQA scores—a state-of-the-art Image Quality Assessment metric. Our method consistently surpasses prior approaches, achieving substantial speedups over full-step generation while maintaining competitive fidelity. Notably, speedup is measured relative to the full 50-step generation. Although InstaFlow and PerFlow (Table 1) are trained on Stable-Diffusion variants—rendering speedup comparisons inexact—the reported latency still provides a meaningful wall-clock comparison.

**Image Editing:** Figure 2 illustrates the performance of our method on image editing using BAGEL and FLUX models. Evaluations are conducted on the GEdit dataset, with GPT-4.1 serving as an automatic judge to score edits on semantic consistency (G_SC), perceptual quality (G_PQ), and an overall quality measure (G_O). Our approach achieves the highest speedup among all baselines while preserving, and in many cases improving, the quality of the edits.

**Video Generation.** Figure 3 reports results on video synthesis using the **VBench** benchmark, which multiple dimensions including motion dynamics, temporal consistency, and scene composition. For perceptual assessment of individual frames, we additionally employ the no-reference **BRISQUE** metric. Our method consistently surpasses baselines, delivering sharper frames and more coherent temporal evolution while achieving substantial acceleration.

Collectively, these results demonstrate that FastFlow offers a transformative trade-off between efficiency and quality. By dramatically reducing computation without sacrificing perceptual or semantic

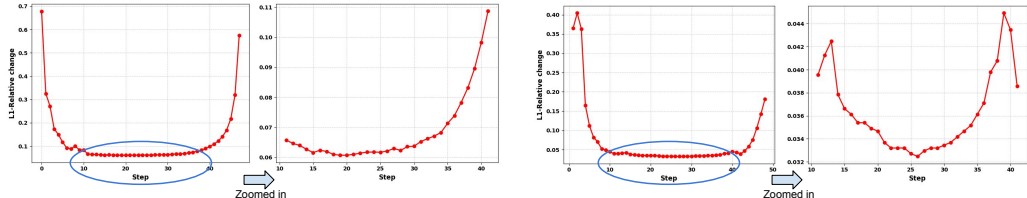

Figure 4: L1-relative error between consecutive velocity predictions in the BAGEL model. While the trajectories may appear constant at intermediate scales, a finer analysis uncovers subtle yet systematic variations, indicating that the underlying dynamics are not strictly stable.

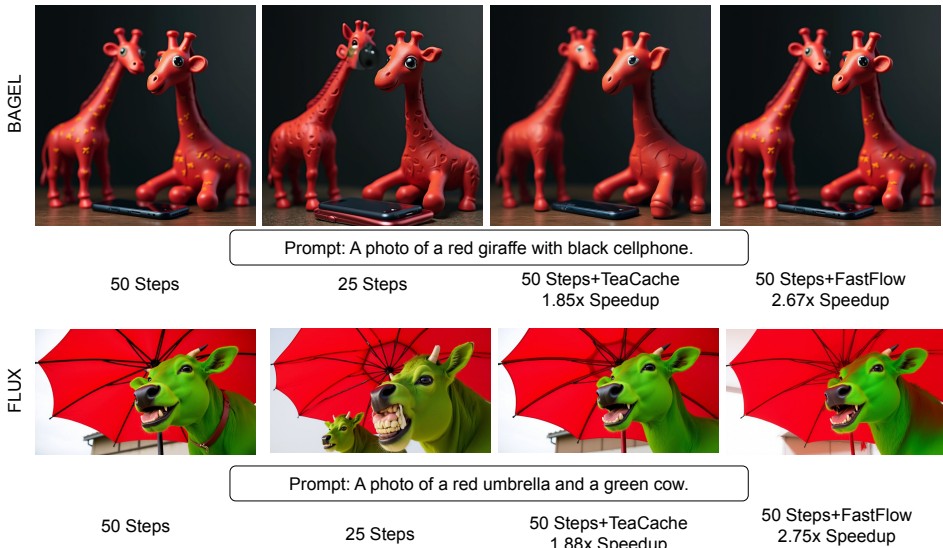

Figure 5: Generated instance for image generation task for BAGEL and FLUX models.

fidelity, our method sets a new standard for fast, high-quality image generation and editing. This opens the door to practical, real-time applications on resource-constrained devices, making advanced generative modeling more accessible and scalable.

## 4.2 ANALYSIS

**Empirical evidence motivating approximation:** In Figure 4, we illustrate the L1-relative error $\frac{||v(x_{t_k}, t_k) - v(x_{t_{k+1}}, t_{k+1})||}{||v(x_{t_{k+1}}, t_{k+1})||}$ of velocity predictions across consecutive steps, providing insight into how the model refines trajectories over time. We observe a clear three-phase pattern: the model first establishes the coarse flow, then performs subtle refinements during intermediate steps, and finally adjusts again in the later steps to finalize the trajectory.

Prior work Liu et al. (2025a) has largely relied on zoomed-out trends, where intermediate updates appear nearly constant, motivating strategies that approximate future states solely from the last step. However, a finer-grained inspection reveals that these intermediate refinements, though small, are not negligible—subtle fluctuations accumulate and can lead to significant deviations if ignored. This phenomenon, consistently observed in both generation and editing tasks, highlights the need for approximation methods that capture intermediate dynamics rather than oversimplifying them.

**Qualitative Analysis.** Figure 5 illustrates image generation with BAGEL and FLUX. Simply truncating steps severely harms fidelity, especially for challenging prompts, yielding incomplete or distorted images. TeaCache produces closer outputs but still misses fine-grained details and realism. In contrast, FastFlow delivers results nearly indistinguishable from full-step generation, while being substantially faster.

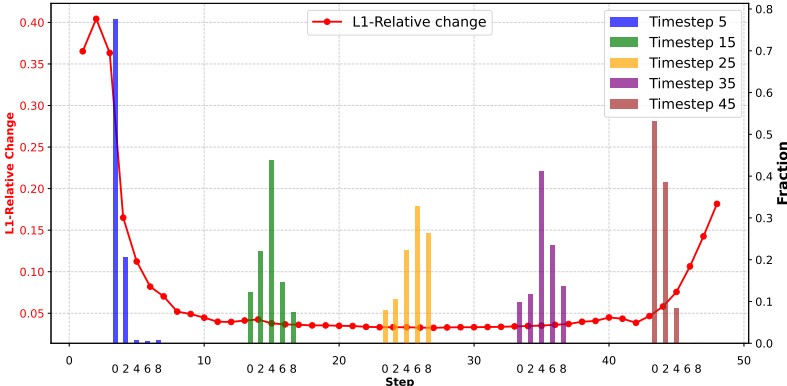

Figure 6: The figure illustrates how our method adaptively skips redundant steps in regions of slow variation, while reverting to full model evaluations when velocity changes are significant. Explanation can be found in Appendix A.3

Figure 10 shows editing examples. Direct step reduction either fails to apply edits or introduces visual artifacts. TeaCache improves but struggles with precise integration. FastFlow, however, incorporates edits similar to full generation, consistently preserving semantic intent and visual quality.

In summary, the strong speedup of FastFlow stems from its aggressive yet principled approximation, which skips redundant updates without diverging from the model trajectory. The multi-armed bandit controller further adapts to dataset and model dynamics (see Figure 11), learning where to skip and where to refine, enabling acceleration without sacrificing quality.

## 5 CONCLUSION

We introduced a new framework FastFlow, that accelerates flow-based generative models by *adaptively skipping redundant steps* while safely approximating the underlying trajectory. Unlike static reduction strategies, our approach dynamically adjusts to the difficulty of incoming samples—skipping more aggressively for easy cases while allocating more model computation to harder ones. The trajectory approximation further empowers the decision-maker (MAB) to capture fine-grained variations, enabling efficient yet faithful generation. Across diverse datasets and tasks, our method consistently outperforms existing baselines, establishing a new paradigm for fast, high-fidelity generative modeling. One limitation of using MAB is the speedup may not be observed in the initial steps due to inherent explorations.

## 6 ETHICS AND REPRODUCIBILITY STATEMENT

We confirm that we have read and adhere to the conference Code of Ethics. To the best of our knowledge, this work is original and all relevant prior work has been properly cited. All authors have contributed to and take responsibility for the content. No LLMs were used in the preparation of this manuscript, except where explicitly disclosed as part of the proposed method.

## ACKNOWLEDGEMENTS

Divya Jyoti Bajpai is supported by the Prime Minister's Research Fellowship (PMRF), Govt. of India. Manjesh K. Hanawal thanks funding support from Telecom Centre of Excellence(TCOE), Department of Telecommunication (DoT) and Ministry of Electronics and Information Technology (MeitY). We also thank funding support from Amazon IIT-Bombay AI-ML Initiative (AIAIMLI).

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

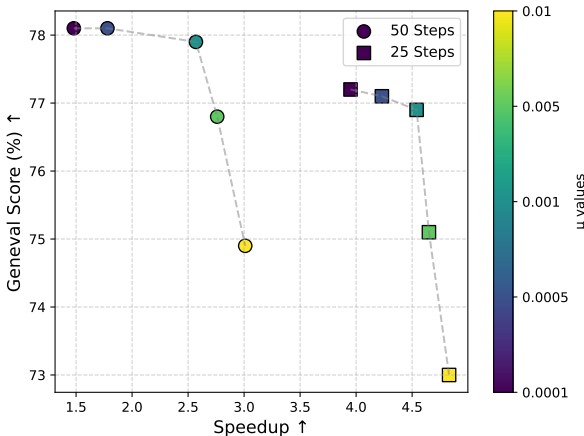

Figure 7: Caption

Ling Yang, Zixiang Zhang, Zhilong Zhang, Xingchao Liu, Minkai Xu, Wentao Zhang, Chenlin Meng, Stefano Ermon, and Bin Cui. Consistency flow matching: Defining straight flows with velocity consistency. *arXiv preprint arXiv:2407.02398*, 2024.

Yuchen Zhang and Jian Zhou. Inverse flow and consistency models. In *Forty-second International Conference on Machine Learning*, 2025.

Yuanzhi Zhu, Zhaohai Li, Tianwei Wang, Mengchao He, and Cong Yao. Conditional text image generation with diffusion models. In *Proceedings of the IEEE/CVF Conference on Computer Vision and Pattern Recognition*, pp. 14235–14245, 2023.

Pascal Zwick, Nils Friederich, Maximilian Beichter, Lennart Hilbert, Ralf Mikut, and Oliver Bringmann. Lediflow: Learned distribution-guided flow matching to accelerate image generation. *arXiv preprint arXiv:2505.20723*, 2025.

# A    APPENDIX

## A.1    FASTFLOW VS. TEACACHE VS. DIRECT REDUCTION

**FastFlow vs. Direct Reduction.** A natural question arises: why not simply reduce the number of diffusion steps directly? While straightforward, this approach is fundamentally different from FastFlow. Direct reduction applies a *static truncation* of steps, which effectively assumes that the velocity at a removed step can be approximated by reusing the velocity from the previous step. This oversimplification discards useful intermediate information and can degrade generation quality.

In contrast, FastFlow is *adaptive*. Instead of discarding steps outright, it selectively identifies which steps must be computed by the model and which can be approximated using prior computations. This ensures that efficiency is achieved without sacrificing fidelity, especially for complex prompts.

**FastFlow vs. TeaCache.** TeaCache approaches acceleration differently: it uses timestep embeddings of noisy inputs and applies a polynomial fitting scheme to determine whether a step should be cached. While conceptually simple, this design comes with two key limitations: 1) It requires calibration via polynomial fitting, which introduces task- and model-specific tuning overhead. 2) It is not truly dynamic — the caching threshold is fixed once a target speedup is specified, and in our empirical evaluation (50 video generations and 100 image generations), we consistently observed that TeaCache converges to repetitive caching patterns (e.g., alternating steps for $2\times$ speedup), regardless of the complexity of the prompt. This indicates that the method does not effectively adapt to prompt-specific generation difficulty, but rather applies a globally fixed caching strategy.

**Advantages of FastFlow.** Unlike TeaCache, FastFlow is *truly dynamic*. It learns on-the-fly during inference and adapts to the complexity of each prompt. For simpler generations, it aggressively

reduces the number of model calls to maximize speedup, while for more complex prompts, it reverts to additional steps to preserve quality. This adaptiveness enables FastFlow to provide consistently better trade-offs between efficiency and fidelity compared to static baselines like TeaCache.

### A.2 SPEEDUP VS PERFORMANCE CURVE

In Figure 7, we show the speedup vs performance trade-off in our method, it also gives a sense of what number of steps to choose and with what $\mu$, given a speedup. For instance, given a speedup of $2.3\times$, whether user should choose 50 steps with some high value of $\mu$ or 25 steps with some low values of $\mu$.

### A.3 FASTFLOW'S SKIP PATTERNS:

Figure 6 illustrates how our method adapts skip decisions to the model's internal dynamics. The plot shows the mean squared error (MSE) of consecutive velocities alongside the frequency with which different skip lengths are selected.

When velocity fluctuations are high, FastFlow consistently chooses shorter skips, preserving accuracy. During intermediate regions where changes are smoother, it shifts toward longer skips, accelerating computation. Finally, as fluctuations re-emerge toward later steps, the method automatically reduces the skip length again. This adaptive behavior explains why FastFlow achieves significant speedups without compromising fidelity: it skips aggressively only when the trajectory is stable and reverts to fine-grained updates when the dynamics demand precision.

### A.4 ADAPTIVENESS OF FASTFLOW

The adaptiveness of our method can be further seen in the Figure 11 we find that our method can adjust the speedup based on the complexity of the incoming samples, if a sample required more model computations it reverts back to lesser number of skips and approximations and in the case of an easier generation the model can further gain speedup by aggressively skipping multiple redundant steps making our method a go to choice for adaptive visual generation.

### A.5 REGRET CURVES

In Figure 8, we plot the cumulative regret of bandits deployed at different points in the denoising trajectory. Across all positions, the cumulative regret begins to flatten between 50 and 100 samples, demonstrating that the bandit rapidly identifies the near-optimal skipping strategy. The small amount of regret accumulated in the first 50 samples is expected because several arms are only marginally suboptimal; choosing them does not significantly affect performance.

A key reason for this fast convergence is the reward structure: it assigns a noticeably higher reward to the truly optimal arms, creating a strong separation between optimal and suboptimal choices. This makes the identification of the best arm easier, allowing the bandit to switch from exploration to exploitation very quickly. As a result, our method converges rapidly within a small number of samples.

### A.6 EMPIRICAL VALIDATION OF ERROR BOUND

In Figure 9, we provide results that verify the tightness of the bound which is developed in Theorem 3.1. The bound clearly shows a good amount of tightness when the number of skipped steps are small and as the number of steps skipped incraeses tehe bound becomes weaker, which is intuitive as well since the bound plays

## B THEORETICAL ANALYSIS

**Assumptions:** The velocity field $v(x, t)$ is assumed to be smooth and satisfy the following conditions for constants $L_x, M > 0$:

1. Lipschitz continuity in space: $\|v(x, t) - v(y, t)\| \leq L_x \|x - y\|$ for all $x, y \in \mathbb{R}^d$.

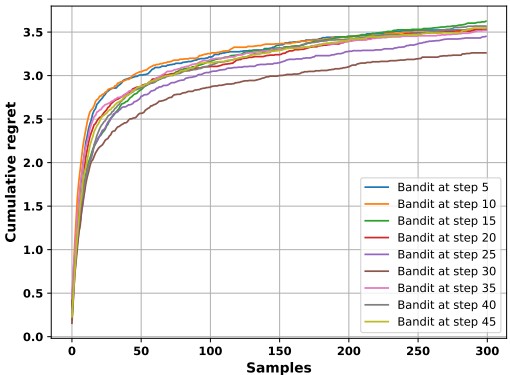

Figure 8: The regret curves for various bandits applied at different steps, the results are the average of five random runs where randomness comes from the reshuffling of data.

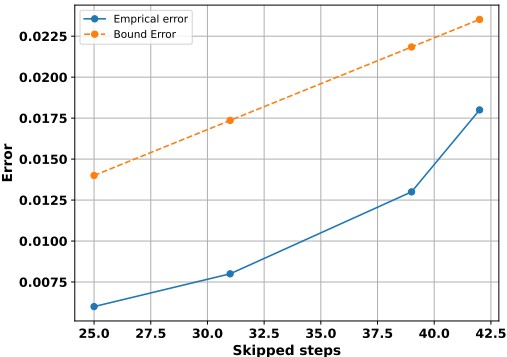

Figure 9: The plot represents empirical error vs the error developed in the Theorem 3.1.

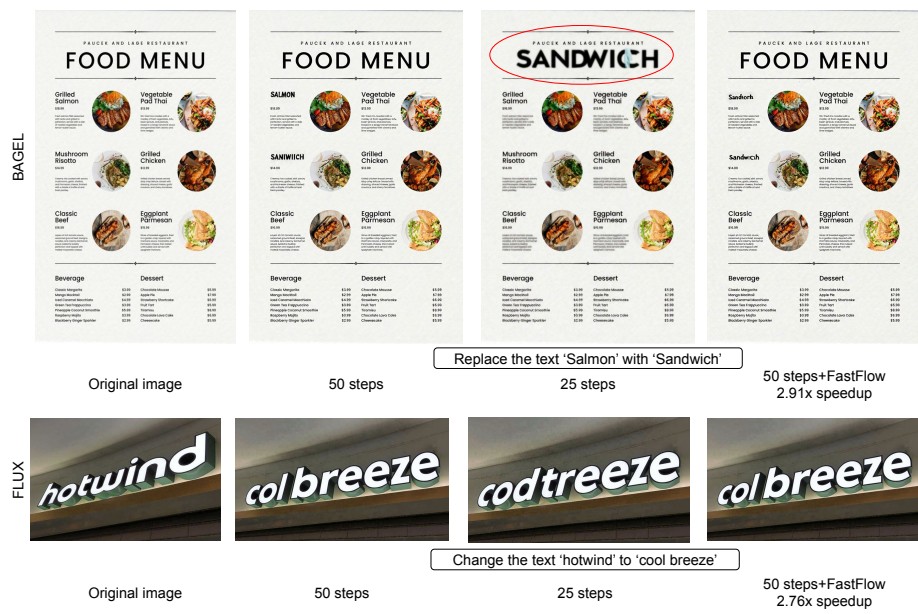

Figure 10: Generated instance for image editing task for BAGEL and FLUX models.

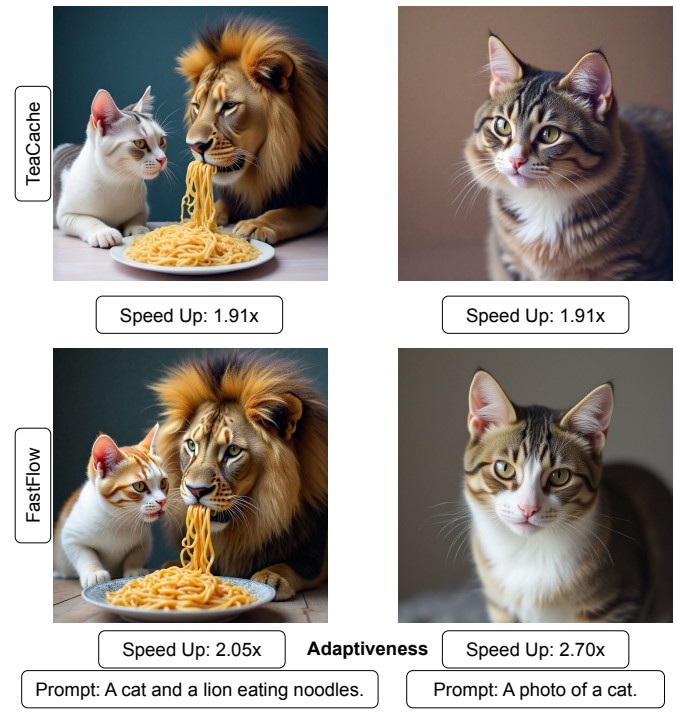

Figure 11: An illustration of the adaptiveness of our method. For simpler generation prompts, it achieves higher speedups by reducing inference calls, whereas for more complex samples, FastFlow automatically reverts to additional model calls. In contrast, baselines such as Teacache remain static across timesteps, showing no dependence on generation complexity.

| Hyperparam | Flux | | BAGEL | |
|---|---|---|---|---|
| | Generation | Editing | Generation | Editing |
| Image resol. | 1360×768 | – | 1024 × 1024 | – |
| Guidance scale | 2.5 | 2.5 | cfg_text_scale = 4 | cfg_text_scale = 4 |
| Guidance rescale | 0 | 0 | cfg_img_scale = 1 | cfg_img_scale = 2 |
| $\mu$ | 0.001 | 0.005 | 0.001 | 0.005 |
| cfg_interval | – | – | [0.4, 1.0] | [0.0, 1.0] |
| timestep_shift | – | – | 3 | 3 |
| cfg_renorm_min | – | – | 0 | 0 |
| cfg_renorm_type | – | – | "global" | "text_channel" |
| Arms | [0,2,4,6] (50), [0,2,4,6] (25), [0,1,2,3] (10) | | same as BAGEL | |

Table 3: **Hyperparameter settings for Flux and BAGEL.** We report separate values for generation and editing. Both models share the same arm configurations, while image resolution and conditioning scales differ.

2. Bounded second time-derivative: $\left\| \frac{\partial^2 v(x,t)}{\partial t^2} \right\| \leq M$ for all $x \in \mathbb{R}^d$.

**Theorem B.1.** *Let $\{x_{t_k}^{\text{true}}\}$ denote the trajectory obtained using the exact velocity field with the forward Euler method, and let $\{x_{t_k}^{\text{approx}}\}$ be the trajectory where velocity evaluations are skipped at a subset of steps $\mathcal{S} \subseteq \{0, \ldots, T-1\}$ and are instead approximated, for simplicity, via a first-order Taylor expansion in time.*

*Under above assumptions, the cumulative error in the final state after $T$ steps with a uniform step size $\Delta t = 1/T$ is bounded by:*

$$e_T := \|x_{t_T}^{\text{approx}} - x_{t_T}^{\text{true}}\| = \mathcal{O}\left(\frac{|\mathcal{S}|}{T^3}\right).$$

*Proof.* Let $e_k := \|x_{t_k}^{\text{approx}} - x_{t_k}^{\text{true}}\|$ be the spatial error at timestep $t_k$. The forward Euler updates for the true and approximate trajectories are given by:

$$x_{t_{k+1}}^{\text{true}} = x_{t_k}^{\text{true}} + \Delta t \cdot v(x_{t_k}^{\text{true}}, t_k)$$
$$x_{t_{k+1}}^{\text{approx}} = x_{t_k}^{\text{approx}} + \Delta t \cdot \tilde{v}_k$$

where $\tilde{v}_k$ is the (potentially approximated) velocity used at step $k$.

By subtracting the two update equations and applying the triangle inequality, we derive the error recurrence:

$$e_{k+1} = \|x_{t_k}^{\text{approx}} - x_{t_k}^{\text{true}} + \Delta t(\tilde{v}_k - v(x_{t_k}^{\text{true}}, t_k))\|$$
$$\leq e_k + \Delta t \cdot \|\tilde{v}_k - v(x_{t_k}^{\text{true}}, t_k)\|.$$

We bound the velocity mismatch term by splitting it into the approximation error and the propagated error:

$$\|\tilde{v}_k - v(x_{t_k}^{\text{true}}, t_k)\| \leq \underbrace{\|\tilde{v}_k - v(x_{t_k}^{\text{approx}}, t_k)\|}_{\text{Approximation Error}} + \underbrace{\|v(x_{t_k}^{\text{approx}}, t_k) - v(x_{t_k}^{\text{true}}, t_k)\|}_{\text{Propagated Error}}.$$

The propagated error is bounded by the spatial Lipschitz condition:

$$\|v(x_{t_k}^{\text{approx}}, t_k) - v(x_{t_k}^{\text{true}}, t_k)\| \leq L_x e_k.$$

For the approximation error, if a skip occurs ($k \in \mathcal{S}$), we use the bound on the second derivative. The remainder term of a first-order Taylor expansion gives:

$$\|\tilde{v}_k - v(x_{t_k}^{\text{approx}}, t_k)\| \leq \frac{M}{2}(\Delta t)^2.$$

If no skip occurs ($k \notin \mathcal{S}$), this error is 0. Combining these bounds, the full error recurrence becomes:

$$e_{k+1} \leq e_k + \Delta t \left(\mathbb{1}_{\{k \in \mathcal{S}\}} \cdot \frac{M}{2}(\Delta t)^2 + L_x e_k\right)$$
$$\leq (1 + L_x \Delta t)e_k + \mathbb{1}_{\{k \in \mathcal{S}\}} \cdot \frac{M}{2}(\Delta t)^3.$$

We unroll this recurrence starting from the initial condition $e_0 = 0$:

$$e_T \leq \sum_{k=0}^{T-1} \mathbb{1}_{\{k \in \mathcal{S}\}} \cdot \frac{M}{2}(\Delta t)^3 \cdot (1 + L_x \Delta t)^{T-1-k}.$$

Using the inequality $(1 + x) \leq e^x$, we can bound the exponential term:

$$(1 + L_x \Delta t)^{T-1-k} \leq e^{L_x \Delta t(T-1-k)} \leq e^{L_x T \Delta t} = e^{L_x}.$$

Substituting this back, we get a sum over the $|\mathcal{S}|$ steps where an error was introduced:

$$e_T \leq \sum_{k \in \mathcal{S}} \frac{M}{2}(\Delta t)^3 \cdot e^{L_x} = |\mathcal{S}| \cdot \frac{M e^{L_x}}{2}(\Delta t)^3.$$

Finally, with $\Delta t = 1/T$, the bound is:

$$e_T \leq \left(\frac{|\mathcal{S}| M e^{L_x}}{2}\right)\frac{1}{T^3},$$

which proves that $e_T = \mathcal{O}\left(\frac{|\mathcal{S}|}{T^3}\right)$. $\qquad\square$

**Interpretation:** The upper bound on the overall error can be interpreted as:

1. We show that global error between our update to the Euler solver, and Euler solver with full evaluation is small (as above).

2. The overall error grows linearly with the number of skipped steps, but decays rapidly with the total number of timesteps.

3. The proof provides a strong theoretical motivation for our choice of the bandit reward. The goal is to maximize the number of skipped steps, while lowering the local error.

4. Approximating velocities in regions of large curvature (i.e., large $M$) increases the overall error.

## C  FASTFLOW AND LINEAR MULTI-STEP METHODS

Given the first-order Taylor approximation for velocity, and Euler solver for FastFlow, a re-formulation to cast our updates in form of a Linear multi-step can be obtained as follows.

Assume that as per our method and a uniform step-size, we evaluate the velocities $v(x_k, k), v(x_{k-1}, k-1)$ and then skip $m$ steps as per a bandit action. Then $x_{k+m+1}$, with a Euler step update is given by:

$$x_{k+m+1} = x_{k+m} + h \cdot v(x_{k+m}, k+m) \tag{6}$$

We approximate $x_{k+m}$ and $v(x_{k+m}, k+m)$ as follows:

$$x_{k+m} \approx x_k + m \cdot h \cdot v(x_k, k) \tag{7}$$

$$v(x_{k+m}, k+m) \approx v(x_k, k) + m \cdot h \frac{v(x_k, k) - v(x_{k-1}, k-1)}{h} \tag{8}$$

Updating $(2), (3)$ in $(1)$, with some simplification results in:

$$x_{k+m+1} = x_k + h[(2m+1) \cdot v(x_k, k) - m \cdot v(x_{k-1}, k-1)] \tag{9}$$

This is an multi-step update with order=1. The update is consistent, and the Local Truncation Error (at step $k+m+1$, when bandit skips $m$ steps at step $k$) can be shown to be as (after accounting for a division by $h$ in Suli & Mayers (2003)):

$$\tau_{k,m} = \frac{m^2+1}{2 \cdot (m+1)} \cdot h^2 \cdot \frac{d^2x}{dt^2}(t_k) + \mathcal{O}(h^3) \leq \frac{m}{2} \cdot h^2 \cdot \frac{d^2x}{dt^2}(t_k) + \mathcal{O}(h^3) \tag{10}$$

### C.1  SOLVER-AGNOSTIC ACCELERATION

Our primary contribution is a *training-free* acceleration framework for Flow Matching models based on a dynamic bandit objective. Unlike heuristic step-skipping strategies, we cast the decision to skip model evaluations as an adaptive bandit optimization problem. The method is solver-agnostic and operates on top of existing numerical solvers without modifying their update rules. It reduces computation by selectively skipping model evaluations during inference and does not introduce a new solver. In the main paper, results are shown with the Euler solver; here we additionally evaluate higher-order solvers.

We report results on the BAGEL model evaluated on the GenEval dataset. Speedup is computed as

$$\text{Speedup} = \frac{T_{\max}}{\text{Average number of model evaluations}},$$

with $T_{\max} = 50$. "Overall" and "CLIPIQA" denote perceptual quality metrics (higher is better).

| Method | SO | TO | CO | CL | ATTR | PO | Overall | CLIPIQA | Speedup |
|---|---|---|---|---|---|---|---|---|---|
| Euler (Full, $T$=50) | 0.99 | 0.90 | 0.81 | 0.85 | 0.59 | 0.54 | 0.78 | 0.85 | 1.00 |
| 2-AB (Full, $T$=50) | 0.99 | 0.93 | 0.81 | 0.87 | 0.60 | 0.48 | 0.78 | 0.85 | 1.00 |
| 4-AB (Full, $T$=50) | 0.99 | 0.92 | 0.80 | 0.87 | 0.61 | 0.48 | 0.78 | 0.87 | 1.00 |
| Euler + FastFlow | 0.99 | 0.91 | 0.80 | 0.86 | 0.63 | 0.51 | 0.78 | 0.83 | 2.65 |
| 2-AB + FastFlow | 0.99 | 0.93 | 0.80 | 0.87 | 0.59 | 0.48 | 0.78 | 0.83 | 2.45 |
| 4-AB + FastFlow | 0.99 | 0.93 | 0.81 | 0.87 | 0.61 | 0.48 | 0.78 | 0.84 | 2.37 |

Table 4: Performance across solvers on GenEval. FastFlow consistently achieves $2.37\times$–$2.65\times$ speedup while maintaining comparable perceptual quality, demonstrating solver-agnostic behavior.

