# OpenReview forum: "FastFlow: Accelerating The Generative Flow Matching Models with Bandit Inference"
_ICLR.cc/2026/Conference — ICLR 2026 Poster_

### Official Review · Reviewer_r4c2 · 2025-10-24

**Soundness:** 3
**Presentation:** 3
**Contribution:** 3
**Rating:** 6
**Confidence:** 4

**Summary:**

The paper proposes FastFlow, a plug-and-play adaptive inference framework that accelerates flow-matching generative models without retraining. The key idea is to approximate redundant denoising steps using a finite-difference Taylor expansion of the model’s velocity field, thereby skipping expensive neural evaluations when the dynamics are locally smooth. To decide how many steps can be safely skipped, FastFlow formulates the process as a multi-armed bandit (MAB) problem that adaptively balances efficiency and fidelity during sampling. Overall, FastFlow achieves acceleration while maintaining perceptual and semantic fidelity across multiple generative domains.

**Strengths:**

The paper is clearly written and easy to follow, with a well-motivated goal: accelerating flow-matching models to benefit the broader generative modeling ecosystem. I especially appreciate the inclusion of image editing, where real-time interaction is critical. While reusing the previous step’s velocity is not new, bandit-driven policy for adaptive step skipping is novel in this context to my knowledge and is presented in a concrete, convincing way.

**Weaknesses:**

The proposed method relies on heuristic parameters such as $p$ for velocity approximation and $\mu$ for the reward regularization term. It would be helpful to clarify how these parameters are chosen in practice and whether the method is robust to variations in their values.

Fig. 2–3 consistently show that FastFlow has higher latency than TeaCache (for comparable compute). Is this overhead coming from the multi-armed bandits? It seems that the overhead is not negligible. Can the authors clarify this behavior? Similarly, the result for FastFlow-10 in image generation (Table 1) shows only minor gains compared to Full-10. Additionally, the authors mention that the baselines follow the official hyperparameters. What are these parameters, and could this violate an apples-to-apples comparison? Overall, my concern is that the experiments either show marginal gains or may have presented in an unfair manner.

Lastly, the paper would be strengthened by including an ablation study on the contribution of the multi-armed bandit algorithm. For instance, comparing FastFlow against simpler alternatives such as uniform or piecewise-constant skipping schedules.

**Questions:**

How does the method perform when coupled with quantization method? Also can this be utilized in flow models after reflow training? Ideally, reflow models have straight, non-crossing paths where the proposed method might not be as effective. I supposed experiment presented in Figure 4 using rectified flow models FLUX schnell can show a different trend.

---

> ### Author Response · Authors · 2025-11-21
> **Rebuttal**
>
> We thank the reviewer for time and efforts in reviewing our work.
>
> **Q1:** The proposed method relies on heuristic parameters, such as p for velocity approximation and mu for the reward regularization term. It would be helpful to clarify how these parameters are chosen in practice and whether the method is robust to variations in their values.
>
> **A1:** We would like to clarify that the parameter $p$ is not a tunable hyperparameter. It is a positional index that refers to the second most recent step at which a full model invocation occurred. More specifically, $p$ identifies the “previous-to-previous” step when the model was invoked.
>
> The parameter $\mu$, on the other hand, serves as a scaling coefficient in the reward regularization term to ensure that the magnitude of the regularization aligns with the step-size of the update. $\mu$ can also serve as a trade-off factor between generation quality and computational speedup. Smaller values of $\mu$ emphasize stability and fidelity to the full model outputs, while larger values promote more aggressive acceleration.
>
> To ensure consistency, we provide an explicit formulation for setting $\mu$ in line 374 of the paper, where it is scaled in proportion to the expected step size and the empirical range of the reward term.
>
> **Q2:** Fig. 2–3 consistently show that FastFlow has higher latency than TeaCache (for comparable compute). Is this overhead coming from the multi-armed bandits? It seems that the overhead is not negligible. Can the authors clarify this behavior?
>
> **A2:** We believe Figures 2 and 3 are misinterpreted. These figures plot Geneval score vs. speedup (not latency). The horizontal axis represents speedup, so methods appearing farther to the right achieve higher speedup. Under this correct interpretation, FastFlow consistently yields greater speedup than TeaCache, demonstrating better efficiency rather than higher latency.
> We will refine the figures and captions to more clearly distinguish between speedup and latency, ensuring this misunderstanding does not occur in future revisions.
>
> In Figure 3, lower BRISQUE scores indicate better quality (hence our plot is below the baselines), whereas all other metrics follow the convention that higher scores correspond to better performance (hence our method is above the baselines).
>
> **Q3:** Similarly, the result for FastFlow-10 in image generation (Table 1) shows only minor gains compared to Full-10.
>
> **A3:** The apparent small difference between FastFlow-10 and Full-10 is primarily due to the reduced opportunity for speedup when the total number of sampling steps is already small. Still, our method finds any redundant steps and speeds up the inference on that.
> For clarity, the Full-10 baseline achieves a 5x speedup relative to the Full-50 setting (50/10 = 5x). FastFlow-10, however, achieves a ~7x speedup with a negligible drop in quality. The additional speedup is achieved through adaptive skipping even in the smaller models.
>
> **Q4:** Additionally, the authors mention that the baselines follow the official hyperparameters. What are these parameters, and could this violate an apples-to-apples comparison? Overall, my concern is that the experiments either show marginal gains or may have presented in an unfair manner.
>
> **A4:** We use three publicly released baselines, InstaFlow, PerFlow, and TeaCache, and follow their official repository without modification. PerFlow provides options to choose the step size for inference as 6, 8, or 10. We set it to 10, as it provided a good trade-off between quality and speedup. In TeaCache, the inference-time threshold (which determines whether a step is recomputed or reused) is a hyperparameter with a default value of 0.6. We used the default setting.
>
> **Fair comparison:** All three baselines are evaluated under the exact inference configurations recommended by their respective authors. None of the baselines has hidden or extra tunable parameters that we omit. We emphasize that we did not modify or retune any baseline hyperparameters in ways that might disadvantage them. All experiments are fully reproducible using the public repositories and official settings.

---

> ### Author Response · Authors · 2025-11-21
> **Rebuttal (Extended)**
>
> **Q5:** Lastly, the paper would be strengthened by including an ablation study on the contribution of the multi-armed bandit algorithm. For instance, comparing FastFlow against simpler alternatives such as uniform or piecewise-constant skipping schedules.
>
> **A5:** We appreciate the reviewer’s suggestion. We conducted an ablation comparing FastFlow with two static skipping strategies, uniform skipping and piecewise skipping, while keeping the total number of skipped steps fixed (25 out of 50, and 13 out of 25). Results are reported in the table below.
>
> | **Method**             | **SO** | **TO** | **CO** | **CL** | **ATTR** | **PO** | **Overall** | **CLIPIQA** | **Speedup** |
> |------------------------|--------|--------|--------|--------|----------|--------|-------------|-------------|-------------|
> | **Full 50**            | 0.99   | 0.90   | 0.81   | 0.85   | 0.59     | 0.54   | 0.78        | 0.85        | 1x           |
> | **Full 25**            | 0.99   | 0.91   | 0.78   | 0.84   | 0.62     | 0.51   | 0.77        | 0.82        | 2x           |
> | **_Different sampling decisions_** |        |        |        |        |          |        |             |             |             |
> | **Full 50 uniform-1**  | 0.99   | 0.89   | 0.78   | 0.80   | 0.56     | 0.50   | 0.74        | 0.76        | 2x           |
> | **Full 25 uniform-1**  | 0.99   | 0.91   | 0.75   | 0.78   | 0.55     | 0.48   | 0.73        | 0.72        | 3.84x           |
> | **Full 50 piece**      | 0.99   | 0.90   | 0.80   | 0.79   | 0.57     | 0.49   | 0.75        | 0.79        | 2x           |
> | **Full 25 piece**      | 0.99   | 0.90   | 0.78   | 0.76   | 0.59     | 0.47   | 0.74        | 0.77        | 3.84x
>
> Uniform-1: deterministically skip every alternate step.
>
> Piecewise: skip primarily in the middle region (e.g., around steps 10 and 40 for 50-step sampling and 5 and 20 for 25 steps sampling), based on the observation that early and late steps are typically more important.
>
> Despite matching the total number of skips, static schedules consistently underperform. This is expected: different samples require different amounts of computation, and a fixed skipping pattern cannot adapt to such variation in sample hardness.
>
> **Q6:** How does the method perform when coupled with the quantization method?
>
> **A6:** FastFlow is fully plug-and-play and makes no assumption about the underlying model's architecture, precision, or parameter format. This means it can be directly applied to FP32, BF16, INT8, or any quantized version of a flow-matching model. Quantization only affects the internal computations of the full model evaluations; the FastFlow procedure itself remains unaffected, as it uses only the model outputs and does not rely on weight-level or architecture-specific properties.
>
> Furthermore, the multi-armed bandit that governs the skip decisions is inherently adaptive. Even if quantization slightly alters the velocity predictions (e.g., through precision loss or rounding effects), the bandit automatically adjusts its exploration-exploitation strategy based on the observed reward. This ensures that FastFlow continues to detect redundant steps reliably under quantized conditions.
>
> **Q7:** Also can this be utilized in flow models after reflow training?
>
> **A7:** Yes, FastFlow can be directly applied to models trained with ReFlow without requiring any modification to the method. Although ReFlow modifies how the flow trajectories are learned during training, by iteratively refining the flow direction and reducing variance across trajectory segments, the inference process for these models still consists of step-wise denoising or velocity integration, which is precisely the setting where FastFlow operates.
>
> FastFlow only relies on the model’s output predictions at each step and identifies steps where the update (i.e., change in predicted velocity) is sufficiently small to safely approximate without invoking the expensive full network. It makes no assumptions about the model architecture, training objective, or trajectory construction, and is entirely agnostic to training mechanisms like standard flow matching, consistency distillation, or ReFlow. As long as the model follows a step-wise denoising or integration process, FastFlow can seamlessly plug in and adapt to the refined trajectories produced by ReFlow.
>
> Finally, the L1-relative error between consecutive velocities, as shown in Figure 4 for general Flow Models, may exhibit a different general trend for ReFlow-based methods (due to their smoother trajectories). Any such smoothness inherently creates opportunities for FastFlow to complement ReFlow by reducing redundant steps and thereby decreasing the number of full forward passes, thus accelerating inference further without sacrificing quality.
>
> We hope that we clarified most of your concerns, please let us know if there are any further questions, else we request you to please reassess the score.

---

### Official Review · Reviewer_Tpv2 · 2025-11-01

**Soundness:** 4
**Presentation:** 3
**Contribution:** 3
**Rating:** 6
**Confidence:** 3

**Summary:**

This paper introduces FastFlow, a plug-and-play adaptive inference framework to accelerate generative flow-matching models by skipping redundant denoising steps. The key insight is that flow-matching generative trajectories are often approximately linear, so intermediate states can be extrapolated cheaply instead of recomputing every step. FastFlow uses a finite-difference (Taylor series) approximation of the model’s velocity field to predict future states, allowing the method to advance multiple steps at zero neural network cost. Crucially, the framework employs a multi-armed bandit (MAB) at each timestep to adaptively decide how many steps to skip before the next full model evaluation. The bandit’s reward balances two objectives: (i) speed (skipping more steps) and (ii) accuracy (penalizing deviation from the true model trajectory). By learning this trade-off online per sample, FastFlow dynamically skips only those steps that would have minimal effect on final output. The approach is model-agnostic (no retraining or extra networks required) and integrates seamlessly into existing flow-matching pipelines.

The paper provides a theoretical bound on the error induced by skipping steps, formulates the skip decision as an online bandit problem, and demonstrates various experiments on text-to-image generation, image editing, and text-to-video generation. Empirically, FastFlow achieves over 2.6× speedup in inference while maintaining output quality comparable to the full model across these tasks. This represents a significant improvement over prior static acceleration methods, which often require retraining or sacrifice fidelity.

Though this method is effective when multiple steps are necessary, there are already many few-step or even one-step models (e.g., distillation or shortcut models) available today, so I am not sure whether this method is truly useful in practical scenarios.

**Strengths:**

Unlike static acceleration schemes, FastFlow adapts to each sample’s complexity. The multi-armed bandit dynamically decides per timestep how many steps to skip, meaning simpler cases automatically run faster while complex cases get more compute. This adaptive inference is novel and ensures no one-size-fits-all schedule, leading to greater robustness across diverse inputs.

The proposed framework is model-agnostic and plug-and-play, so it can be applied to existing pretrained flow-matching models without any retraining or fine-tuning. There’s no need for distilling a new model or training an auxiliary network, which makes the method very practical. It can be integrated into current pipelines with minimal effort, offering immediate speed benefits.

**Weaknesses:**

It requires some exploration to learn the optimal skipping policy. You acknowledge that the speedup may not fully materialize in the very first steps or first few samples due to this exploration phase. In practice, you mitigate this by seeding the bandit with one full generation, but if a user only generates a handful of samples, the adaptive policy might not have time to reach peak efficiency. In scenarios with very few inference runs, the benefit of FastFlow could be less pronounced.

The effectiveness of FastFlow rests on the assumption that the generative trajectories are locally smooth/linear enough to be extrapolated. While flow-matching models do encourage linear paths, there might be cases of highly non-linear or complex dynamics where the Taylor approximation could be less accurate. Therefore, essentially, FastFlow may be less effective if the model’s velocity field changes rapidly in unpredictable ways.

Table 1 and Table 2 are overlapped. Please adjust the margin via \vspace.

This paper contains some typos and grammatical issues. Here are the ones I found just by skimming through it:
* L67: a a theoretical -> a theoretical
* L95: We setup -> We set up
* L170: a static criteria -> a static criterion
* L364: is applied is as -> is applied as
* L388: an the -> and the
* L490: it’s content -> its content

**Questions:**

How many samples or iterations does it typically take for the bandit policy to stabilize? In your experiments, after seeding with one full generation, does FastFlow achieve near-optimal skipping immediately on the next sample, or does it require a few generations to fully adapt? Clarifying this can help understand use-cases. Any insight into how the bandit’s learning curve looks would be helpful.

Did you observe any failure cases or significantly reduced speedups for particular input types or prompts that might cause non-linear dynamics? Analyzing a case where FastFlow nearly defaults to the full model would illustrate its limits and robustness.

Theorem 3.1 gives an error bound $O(|S|/T^3)$. Did you empirically measure how close the practical error comes to this bound? In other words, is the bound reasonably tight or very conservative? Some intuition or experiment on how the final output error grows with number of skips in practice.

---

> ### Author Response · Authors · 2025-11-21
> **Rebuttal**
>
> We thank the reviewer for time and efforts in reviewing our work.
>
> **Q1:** It requires some exploration to learn the optimal skipping policy. You acknowledge that the speedup may not fully materialize in the very first steps or first few samples due to this exploration phase. In practice, you mitigate this by seeding the bandit with one full generation, but if a user only generates a handful of samples, the adaptive policy might not have time to reach peak efficiency. In scenarios with very few inference runs, the benefit of FastFlow could be less pronounced.
>
> **A1:** We appreciate the reviewer’s concern regarding the exploration phase. It is true that FastFlow involves an initial exploration step to learn optimal skipping patterns, and this phase is fundamental in any bandit setup. The duration of this phase depends on the hardness of the problem. It turns out that this phase is smaller in FastFlow. This is because flow-matching models exhibit consistent denoising trajectories across samples, resulting in skip patterns that generalize quickly across generations for a given model and dataset.
>
> We would like to highlight that the speedups reported in our experiments already account for the exploration phase. We did not exclude or adjust for these initial rounds; the performance numbers reflect full, realistic end-to-end inference time. Even with exploration included, FastFlow consistently outperforms all baselines across every evaluated task.
> Finally, generative models with flow-matching architectures are commonly deployed in long-running scenarios, where large numbers of samples are produced. In such settings, the minor initial exploration cost becomes negligible, while the cumulative computational savings are significant. Thus, even with an inherent exploration period, FastFlow delivers meaningful and reliable speedup from very early in the deployment cycle.
> In the revised version, we have added a cumulative regret plot for bandits at different steps that further clarifies the doubt (see Figure 7 and Appendix A.5).
>
> **Q2:** The effectiveness of FastFlow rests on the assumption that the generative trajectories are locally smooth/linear enough to be extrapolated. While flow-matching models do encourage linear paths, there might be cases of highly non-linear or complex dynamics where the Taylor approximation could be less accurate. Therefore, essentially, FastFlow may be less effective if the model’s velocity field changes rapidly in unpredictable ways.
>
> **A2:** We agree that FastFlow relies on the assumption that the generative dynamics are locally smooth enough for Taylor-based extrapolation. However, we mitigate this concern through two mechanisms:
> Adaptive Bandit Behavior in Complex Regions: We precisely designed the multi-armed bandit mechanism to be sensitive to such situations where the velocity field fluctuates. As shown in Figure 11 (where we plot histogram for the count of various arms at different steps), the bandit rarely suggests skipping during the initial and final phases of the flow, where the velocity tends to be more chaotic (see Figure 4 and 11) or less predictable. This indicates that the bandit learns to avoid skipping in regimes where the Taylor approximation may be less reliable. In other words, when the velocity field becomes unstable and local linearity breaks down, the bandit naturally adapts by reducing or stopping skipping, thereby ensuring robust performance.
>
> Flow Matching Encourages Straight Trajectories: Flow matching models are trained with objectives that explicitly promote straight, consistent generative trajectories. The transport cost minimization inherently aligns the learned velocity field along predictable paths. This built-in structure complements FastFlow’s Taylor approximation in most regions of the inference trajectory.
> Together, these two components, adaptive step skipping and structurally straight flow trajectories, enable FastFlow to maintain speedups without sacrificing generation quality, even when confronted with somewhat non-linear dynamics.

---

> ### Author Response · Authors · 2025-11-21
> **Rebuttal (Extended)**
>
> **Q3:** Did you observe any failure cases or significantly reduced speedups for particular input types or prompts that might cause non-linear dynamics? Analyzing a case where FastFlow nearly defaults to the full model would illustrate its limits and robustness.
>
> **A3:** We did not observe explicit failure cases for FastFlow in our experiments; However, we observed that for complex prompts, the speedup lowers, and for simpler prompts, the model aggressively skips multiple steps. Such a behaviour of our method is shown in Figure 10.
>
> As for input-dependent behavior, we did not find any specific prompt types or input complexities that caused the method to revert to the full generation process. Across a diverse range of prompts and datasets, FastFlow consistently adapted its skip strategy based on the observed local dynamics, ensuring that steps exhibiting non-linear behavior were not skipped. This input-aware adaptivity explains why the method performs reliably without introducing failure cases attributable to prompt characteristics.
>
>
>
> **Q4:** Theorem 3.1 gives an error bound . Did you empirically measure how close the practical error comes to this bound? In other words, is the bound reasonably tight or very conservative? Some intuition or experiment on how the final output error grows with number of skips in practice.
>
> **A4:** We empirically evaluated the tightness of Theorem 3.1 by comparing the empirical error vs the error bound found in Theorem 3.1 across different skip budgets. The results are now included in the paper in Figure 8.
> We observe that when only a small number of steps are skipped, the empirical error is substantially lower than the theoretical bound, indicating that the bound is conservative in this regime. As the number of skipped steps increases, the empirical error grows accordingly and approaches the theoretical curve. However, across all skip configurations we tested, the empirical error consistently remained below the bound, confirming that Theorem 3.1 provides a valid and reliable worst-case guarantee.
>
> We hope that we clarified most of your concerns, please let us know if there are any further questions, else, we request you to please reassess the score.

---

### Official Review · Reviewer_qEHL · 2025-11-01

**Soundness:** 3
**Presentation:** 3
**Contribution:** 3
**Rating:** 6
**Confidence:** 3

**Summary:**

This paper proposes FastFlow, a plug-and-play adaptive inference framework that accelerates generation in flow matching models. FastFlow identifies denoising steps that produce only minor adjustments to the denoising path and approximates them without using the full neural network models used for velocity predictions. The approximation utilizes finite-difference velocity estimates from prior predictions to efficiently extrapolate future states, enabling faster advancements along the denoising path at zero compute cost.

**Strengths:**

- the motivation to accelerate flow-matching models is reasonable.
- the proposed method is plug-and-play and introduce negalectable extra costs

**Weaknesses:**

- missing comparisons on ImageNet 256

**Questions:**

can the method combined with modern fast samplers instead of Euler?

---

> ### Author Response · Authors · 2025-11-21
> **Rebuttal**
>
> We thank the reviewer for time and efforts in reviewing our work.
>
> **Q1:** missing comparisons on ImageNet 256.
>
> **A1:** We did not include comparisons on ImageNet 256 because the underlying models used in our experiments are designed specifically for text-to-image and text-to-video generation, and do not support class-conditional sampling in the format required for ImageNet-style evaluation.
>
> Instead, we evaluate our method on recently established and comprehensive benchmarks such as Geneval, GEdit, and V-Bench. These datasets cover a much broader range of generation tasks and are designed to evaluate multiple dimensions of generative quality, including attributes like spatial positioning, color fidelity, object counting, compositional reasoning, and textual alignment, under open-ended text prompts which are not covered in imagenet-256. This better reflects the operational settings where flow-matching models are typically deployed.
>
> Furthermore, simple class-conditional prompts (e.g., “a photo of a cat,” “a photo of a car”) are already implicitly covered within these datasets. However, our chosen benchmarks go beyond class labels to evaluate more realistic and challenging generation scenarios. We believe this aligns more closely with the intended use cases of the models and more rigorously tests the generalization capabilities of FastFlow.
>
> **Q2:** can the method combined with modern fast samplers instead of Euler?
>
> **A2:** Yes, the method can be extended to other solvers as well. To verify our claim, we implement our method on a two-step and a four-step Adams-Bashforth style solver. The Table below provides the results:
>
> | **Method/Metric**                       | **SO** | **TO** | **CO** | **CL** | **ATTR** | **PO** | **Overall** | **CLIPIQA** | **Speedup** |
> |-----------------------------------------|--------|--------|--------|--------|----------|--------|-------------|-------------|-------------|
> | **Euler (Full, T=50)**                  | 0.99   | 0.90   | 0.81   | 0.85   | 0.59     | 0.54   | 0.78        | 0.85        | 1x           |
> | **2-Adams-Bashforth (Full, T=50)**      | 0.99   | 0.93   | 0.81   | 0.87   | 0.60     | 0.48   | 0.78        | 0.85        | 1x           |
> | **4-Adams-Bashforth (Full, T=50)**      | 0.99   | 0.92   | 0.80   | 0.87   | 0.61     | 0.48   | 0.78        | 0.87        | 1x          |
> | **_FastFlow_**                          |        |        |        |        |          |        |             |             |             |
> | **FastFlow (T=50)**                     | 0.99   | 0.91   | 0.80   | 0.86   | 0.63     | 0.51   | 0.78        | 0.83        | 2.65x        |
> | **_Adams-Bashforth with FastFlow_**     |        |        |        |        |          |        |             |             |             |
> | **2-AB + FastFlow (T=50)**              | 0.99   | 0.93   | 0.80   | 0.87   | 0.59     | 0.48   | 0.78        | 0.83        | 2.45x        |
> | **4-AB + FastFlow (T=50)**              | 0.99   | 0.93   | 0.81   | 0.87   | 0.61     | 0.48   | 0.78        | 0.84        | 2.37x        |
>
> We hope that we clarified most of your concerns, please let us know if you have any further questions, else, we request you to please reassess the score.

---

### Official Review · Reviewer_D8Ff · 2025-11-03

**Soundness:** 3
**Presentation:** 2
**Contribution:** 2
**Rating:** 4
**Confidence:** 4

**Summary:**

This paper proposes a training free, plug and play acceleration scheme for flow matching Euler sampler at inference time using multi-arm bandit algorithms to choose the most relevant steps given a low-budget step sizes. More specifically, FastFlow aims to skip a variable number of intermediate time steps and approximate the missing velocities using a finite difference in time, and the choice of how many steps to skip is cast as a multiarmed bandit with reward. A theorem gives a bound on the terminal deviation between the approximated and full trajectories under smoothness assumptions, with uniform step size and a set $\mathcal{S}$ of skipped steps. Experiments on image generation, image editing, and video generation claim speedups up to about 2.6 times while maintaining GenEval and CLIP based IQA metrics near full sampling. Qualitative examples are shown for BAGEL and FLUX models and HunyuanVideo.

**Strengths:**

I think the most notable point is that the method is training-free and easy to integrate into existing flow matching pipelines. The speedup figures are plausible given the cost model of flow samplers where every velocity evaluation dominates wall time. I also like recasting the step-size selection as a bandit objective, whichh directly encodes the speed-accuracy tradeoff

**Weaknesses:**

- I think the novelty is thinner than the paper suggests. The velocity extrapolator collapses to a two step Adams Bashforth style predictor in uniform time-step. The work should at least acknowledge this equivalence and position itself relative to, for example,  PNDM [1], and other linear multistep sampling strategies already common in diffusion code bases. Empirically, a direct comparison to a simple two step predictor that still evaluates the model at checkpoints would be informative. Moreover, the statement that most alternative accelerators require retraining is not fully accurate. TeaCache and DeepCache are training free, and the recent adaptive skipping line is also training free in some variants. These should be acknowledged and compared.

- The theory is reassuring but optimistic in scale. For example, with $T=50$ and $∣S∣=25$, the error upper bound term $O(|S|/T^3)$ suggests very small terminal deviations unless the bounding constants are large. However, in the empirical evaluation, the experiments do show quality drop at aggressive skip levels, so either the constants are large or the bound does not capture the dominant error channel. A local error monitor beyond the velocity mismatch would be more principled, for example, an embedded predictor-corrector or curvature proxy, as in adaptive time-stepping literature.

- The experimental section omits several highly related baselines. AdaptiveDiffusion and AdaDiff are the most obvious, but there are also solver learning baselines such as Bespoke Solvers and S4S that reduce NFE without training the base generator. A comparison would help position FastFlow on the quality versus NFE Pareto.


[1] Luping Liu, Yi Ren, Zhijie Lin, Zhou Zhao (2022); Pseudo Numerical Methods for Diffusion Models on Manifolds, ICLR 2022.

[2] Neta Shaul, Juan Perez, Ricky T. Q. Chen, Ali Thabet, Albert Pumarola, Yaron Lipman (2023), Bespoke Solvers for Generative Flow Models, ICLR 2024.

**Questions:**

- Please quantify compute precisely. Report average number of model calls per sample and the distribution of skip lengths $\alpha_t$.
- Please compare against AdaptiveDiffusion and AdaDiff under the same backbones and prompts, and include DeepCache on image tasks and TeaCache on both image and video. Use the same target speed levels and report NFE matched comparisons.
- See also other remarks in Weaknesses on the theoretical bound.

---

> ### Author Response · Authors · 2025-11-21
> **Rebuttal**
>
> We thank the reviewer for time and efforts in reviewing our work.
>
> **Q1:**  The velocity extrapolator collapses to a two step Adams Bashforth style predictor in uniform time-step.
>
> **A1:** It is true that velocity expansion using Taylor series and a finite difference approximation are an obvious step compared to other methods, such as TeaCache, where previous estimates are reused. In light of this, we have updated our paper to present the contributions in a precise but conservative manner.
> In order to place our method in comparison with Linear multi-step solvers like Adams-Bashforth, the following is one local update from our method:
>
> $$x_{k+m+1} = x_{k} + h[(2m+1)\cdot v(x_k,k) - m\cdot v(x_{k-1},k-1)]$$
>
> Above assumes a uniform step-size, and we evaluate the velocities $v(x_k,k)$ and $v(x_{k-1},k-1)$. Then, we skip $m$ steps as per a bandit action. This result is based on Taylor approximation for velocity updates, and an Euler solver for flow $x_k$. As expected, this is a multi-step linear solver with order=$1$. Hence, the method carries out adaptive step-skipping with a simple solver.
>
> **Q2:** Empirically, a direct comparison to a simple two-step predictor that still evaluates the model at checkpoints would be informative."
>
> **A2:** We compare our method with two and four-step Adams-Bashforth under full evaluation (as recommended) and present our results below, which show the performance with higher-order solvers on the BAGEL model and GenEval dataset.
>
> | **Method/Metric**                       | **SO** | **TO** | **CO** | **CL** | **ATTR** | **PO** | **Overall** | **CLIPIQA** | **Speedup** |
> |-----------------------------------------|--------|--------|--------|--------|----------|--------|-------------|-------------|-------------|
> | **Euler (Full, T=50)**                  | 0.99   | 0.90   | 0.81   | 0.85   | 0.59     | 0.54   | 0.78        | 0.85        | 1x           |
> | **2-Adams-Bashforth (Full, T=50)**      | 0.99   | 0.93   | 0.81   | 0.87   | 0.60     | 0.48   | 0.78        | 0.85        | 1x           |
> | **4-Adams-Bashforth (Full, T=50)**      | 0.99   | 0.92   | 0.80   | 0.87   | 0.61     | 0.48   | 0.78        | 0.87        | 1x           |
> | **_FastFlow_**                          |        |        |        |        |          |        |             |             |             |
> | **FastFlow (T=50)**                     | 0.99   | 0.91   | 0.80   | 0.86   | 0.63     | 0.51   | 0.78        | 0.83        | 2.65x        |
> | **_Adams-Bashforth with FastFlow_**     |        |        |        |        |          |        |             |             |             |
> | **2-AB + FastFlow (T=50)**              | 0.99   | 0.93   | 0.80   | 0.87   | 0.59     | 0.48   | 0.78        | 0.83        | 2.45x        |
> | **4-AB + FastFlow (T=50)**              | 0.99   | 0.93   | 0.81   | 0.87   | 0.61     | 0.48   | 0.78        | 0.84        | 2.37x        |
>
> We observe that FastFlow provides speedup without considerable loss in perceptual quality. For the Flow models under consideration, higher-order solvers do not give an improvement in quality which can then be traded off.
>
> However, we show that FastFlow can also be used with these solvers, with the bandit objective providing a trade-off between speed and quality. Specifically, we use these solvers along with steps (for true velocity evaluation) being skipped by bandits. Velocity at the skipped steps is approximated using first-order Taylor series, as in our original method.
>
> **Q3:** The work should at least acknowledge this equivalence and position itself relative to, for example, PNDM [1], and other linear multistep sampling strategies already common in diffusion code bases.
>
> **A3:** We have made updates to acknowledge and comment on the works like PNDM and BeSpoke solvers, which speed up Diffusion models. However, please note that the intent of our method is to work on top of existing solvers and provide potential speed-up without compromising the quality (in practice) of the final output. We do not claim to present a new solver which is fast and provides a better error bound with respect to the underlying (true) flow.
>
> PNDM uses fourth-order Adams-Bashforth in the bulk of its trajectory. As mentioned above, our method results in a simple multi-step solver.
>
> BeSpoke and S4S solvers are specifically trained for given models, with the objective of minimizing the local truncation error. This makes these solvers conditioned on the models. The warm-up of our bandits with the initial few samples may be seen as a training phase comparable to BeSpoke solver training. However, we do not incur additional parameters (except bandit skip lengths set). Additionally, we do not speculate on how BeSpoke and S4S solvers would generalize to Text-guided generation, where complex text prompts may alter the nature of the trajectory for generation.

---

> ### Author Response · Authors · 2025-11-21
> **Rebuttal (Extended)**
>
> **Q4:** TeaCache and DeepCache are training free, and the recent adaptive skipping line is also training free in some variants.
>
> **A4:** TeaCache re-uses previous velocity estimates till a hand-crafted criterion is violated. Results for comparison with TeaCache in image and video are presented in the paper. We also observed that for some given target number of steps, TeaCache reverts to a specific repeating skip schedule.
>
> To the best of our understanding, DeepCache does not generalize well to the models used in our work because it is fundamentally tailored to UNet-style architectures. Its caching mechanism relies on bypassing specific encoder-decoder computations by reusing feature maps through skip connections between the downsampling and upsampling paths. This optimization is tightly coupled to the structural symmetry of UNets and does not directly transfer to architectures that lack such paired stages or use transformer-based backbones.
>
> **Q5:** The theory is reassuring but optimistic in scale.
>
> **A5:** For clarity, we would like to mention that the error bound presented in the theorem is between the error between a "Euler" solver ($x^{true}$)  for flow (with velocity evaluation at all steps) and the approximations that results due to our method (step-skipping and velocity approximation). This error bound does not include the error of the method with respect to true flow trajectory.
>
> However, continuing from our first response on two-step Adams Bashforth, we use numerical analysis literature to compute a Local Truncation Error (with respect to true flow trajectory), $\tau_{k,m}$ when the bandit at step $k$ decides to skip $m$ steps:
>
> $$\tau_{k,m} = \frac{m^2 + 1}{2 \cdot (m+1)}\cdot h^2\cdot \frac{d^2 x}{d t^2}(t_k) + O(h^3) \leq \frac{m}{2}\cdot h^2\cdot \frac{d^2 x}{d t^2}(t_k) + O(h^3)$$
>
> Hence, our method exhibits an error which is similar to an Euler solver, but scales by $m$. This is where the bandit responds to balance between $m$ and local curvature $\frac{d^2x}{dt^2}$.
>
> With respect to empirical results, we would like to mention two important points:
>
> 1) The reported speed-up and quality includes the warm-up samples, used by bandit for exploration. In this phase, the bandits may be sub-optimal, resulting in lower speed-up and quality (thus higher error as compared to theory). These samples are included in our computation for a fair comparison with other methods.
>
> 2) The theorem assumes a fixed uniform step size. However, a few flow models are also trained on non-uniform step sizes.
>
> **Q6:**  Report average number of model calls per sample and the distribution of skip lengths $\alpha_t$.
>
> **A6:** We report speed-up as the ratio of maximum possible evaluations $T$ for a given generation to the average number of model evaluations when our method is applied. One can directly compute the average number of model calls from speedup, more specifically,
>
> Speedup = $\frac{\text{maximum possible evaluations } T}{\text{average number of model evaluations}} $
>
> Representative histograms for the number of skips is shown in Figure 11. The plot in the figure shows the L1-relative error between subsequent generated samples in a sampling trajectory. For the bandits where this error is low (intermediate regions), it can be seen that the corresponding bandits do choose a higher number of skips, while the bandits at the extremes of the trajectory largely decide not to skip any steps.

---

> ### Author Response · Authors · 2025-11-21
> **Rebuttal (Extended)**
>
> **Q7:** Please compare against AdaptiveDiffusion and AdaDiff under the same backbones and prompts, and include DeepCache on image tasks and TeaCache on both image and video.
>
> **A7:** We will updated our results to compare with TeaCache, AdaptiveDiffusion and AdaDiff. For your perusal, same results are also posted below.  All speed-ups are provided as ratio of maximum number of full evaluations ($T$) to the actual evaluations incurred by the method.
>
> While AdaDiff gains more speedup but has a higher hit, both in terms of image quality as well as GenEval score. AdaptiveDiffusion has a lower speedup but good quality. In the given Table below, we provide the results from the AdaDiff and AdaptiveDiffusion models, where both of them have been implemented on our method. While AdaDiff gains more speedup but has a higher hit, both in terms of image quality as well as GenEval score. Whereas AdaptiveDiffusion has a lower speedup but good quality.
>
> | **Method**              | **SO** | **TO** | **CT** | **CL** | **ATTR** | **PO** | **Overall** | **CLIPIQA** | **Spd.** | **Lat.** |
> |-------------------------|--------|--------|--------|--------|----------|--------|-------------|-------------|----------|----------|
> | **AdaDiff (Requires a set of T)**             | 0.99   | 0.89   | 0.76   | 0.81   | 0.54     | 0.48   | 0.74        | 0.73        | 4.25x     | 9.7      |
> | **AdaptiveDiffusion (T = 50)**   | 0.99   | 0.90   | 0.79   | 0.83   | 0.57     | 0.52   | 0.77        | 0.80        | 1.79x     | 21.9     |
> | **TeaCache (T = 50)**            | 0.99   | 0.89   | 0.78   | 0.83   | 0.58     | 0.52   | 0.76        | 0.80        | 1.85x     | 20.6     |
> | **FastFlow (T=50)**     | 0.99   | 0.91   | 0.80   | 0.86   | 0.63     | 0.51   | 0.78    | 0.83    | 2.65x     | 13.7     |
>
>
> We hope that we clarified most of your concerns, if you have any further questions, please let us know, else we request you to please reassess the score.

---

### Author Response · Authors · 2025-11-25
**Gentle Reminder**

Dear Reviewers,

It is a gentle reminder to kindly acknowledge our rebuttal. Let us know if there are any further questions, and if not, we request you to please reassess the scores.

---

> ### Author Response · Authors · 2025-11-26
> **Gentle Reminder**
>
> Dear Reviewers,
>
> It is a gentle reminder to kindly acknowledge our rebuttal. Let us know if there are any further questions, and if not, we request you to please reassess the scores.

---

### Meta-Review · Area_Chair_J7s9 · 2026-01-06

**Summary:**

- FastFlow is a training-free, plug-and-play acceleration for flow-matching inference that skips redundant steps and approximates missing velocities via finite-difference, with a multi-armed bandit choosing how many steps to skip.
- Discussion focused on thin novelty, missing related baselines, and clarifying compute reporting and theory vs practice.
- Rebuttal adds key ablations and clarifies framing; remaining issues are mostly positioning/presentation.

**Reviewer Concerns:**

Concerns addressed:
- Novelty: authors acknowledge the update is a multi-step linear solver (order=2) under uniform steps and reframe contributions more conservatively.
- Comparisons requested: add 2- and 4-step Adams–Bashforth, showing speedup without large quality loss.
- Exploration: reported speedups include warm-up, add cumulative regret evidence; show speedup varies with prompt complexity.

Remaining concerns:
- Broader positioning vs solver-learning baselines remains mostly discussion-level (authors argue those are trained/conditioned).
- Minor presentation issues (figures/typos) to clean up.

**Reviewer Scores:**

- D8Ff is likely to raise their score to 6, core baseline concerns addressed for the most part
- Others likely to stay at 6

---

### Decision · Program_Chairs · 2026-01-26

Accept (Poster)